# Utility of Endoscopic Ultrasound-Guided Fine-Needle Aspiration and Biopsy for Histological Diagnosis of Type 2 Autoimmune Pancreatitis

**DOI:** 10.3390/diagnostics12102464

**Published:** 2022-10-12

**Authors:** Hidehiro Hayashi, Shin Miura, Fumiyoshi Fujishima, Shimpei Kuniyoshi, Kiyoshi Kume, Kazuhiro Kikuta, Shin Hamada, Tetsuya Takikawa, Ryotaro Matsumoto, Mio Ikeda, Takanori Sano, Fumiya Kataoka, Akira Sasaki, Misako Sakano, Atsushi Masamune

**Affiliations:** 1Division of Gastroenterology, Tohoku University Graduate School of Medicine, Sendai 980-8574, Japan; 2Department of Pathology, Tohoku University Hospital, Sendai 980-8574, Japan

**Keywords:** granulocytic epithelial lesions, main pancreatic duct narrowing, inflammatory bowel disease, ulcerative colitis, International Consensus Diagnostic Criteria

## Abstract

In Japan, type 1 autoimmune pancreatitis (AIP) is the most common type of AIP; type 2 AIP is rare. The aim of this study was to clarify the usefulness of endoscopic ultrasound-guided fine-needle aspiration and biopsy (EUS-FNAB) for the diagnosis of type 2 AIP. We analyzed the tissue specimens of 10 patients with suspected type 2 AIP who underwent EUS-FNAB at our hospital between April 2009 and March 2021 for tissue volume and histopathological diagnostic performance. The male-to-female ratio of the patients was 8:2, and the patient age (mean ± standard deviation) was 35.6 ± 15.5 years. EUS-FNAB provided sufficient tissue volume, with high-power field >10 in eight patients (80.0%). Based on the International Consensus Diagnostic Criteria (ICDC), four patients (40.0%) had histological findings corresponding to ICDC level 1, and five patients (50.0%) had histological findings corresponding to ICDC level 2. The results of this study show that EUS-FNB can be considered an alternative method to resection and core-needle biopsy for the collection of tissue samples of type 2 AIP.

## 1. Introduction

Autoimmune pancreatitis (AIP) is a form of pancreatitis caused by abnormal autoimmune mechanisms. AIP is classified into type 1 AIP (an immunoglobulin [Ig] G4-related disease) and type 2 AIP [1,2,3]. The most common type of AIP in East Asia (including Japan) is type 1 AIP; type 2 AIP is rare in East Asia [4]. Type 1 AIP is characterized by lymphoplasmacytic sclerosing pancreatitis, pancreatic swelling, pancreatic duct narrowing, obliterative phlebitis and IgG4-positive plasma cell infiltration [5,6]. In contrast, type 2 AIP is characterized by idiopathic duct-centric chronic pancreatitis, which is histo-pathologically represented by granulocytic epithelial lesions (GELs) [5,6]. Type 2 AIP has an earlier onset than type 1 AIP and is often complicated by inflammatory bowel disease, particularly ulcerative colitis (UC) [7].

The International Consensus Diagnostic Criteria (ICDC) confer a diagnosis of AIP based on adequate pancreatic tissue samples such as core-needle biopsy specimens and resection specimens, and endoscopic ultrasound-guided fine-needle biopsy (EUS-FNAB) is not recommended for the diagnosis of AIP [1,8].

However, with the recent development of new devices such as Franseen needle and fork-tip needle, EUS-FNB can be used to obtain sufficient tissue samples [9,10,11,12].

In fact, the diagnosis of type 1 AIP using EUS-FNB has been reported in many studies, suggesting that EUS-FNAB is useful for the diagnosis of type 1 AIP [13,14,15,16,17]. In contrast, diagnosis of type 2 AIP using EUS-FNAB has been reported in only a few studies. Only eight cases of type 2 AIP diagnosed using EUS-FNAB have been reported in five studies [14,18,19,20,21].

In this study, we reviewed cases of suspected type 2 AIP based on EUS-FNAB performed in our hospital.

## 2. Materials and Methods

Ten patients with ICDC level 1 or level 2 imaging findings and suspected type 2 AIP who underwent EUS-FNAB at our hospital between April 2009 and March 2021 were included in this study. The diagnosis of AIP was based on the ICDC [1]. Pancreatic duct narrowing was evaluated using endoscopic retrograde cholangiopancreatography (ERCP), but in cases where ERCP was not performed, it was evaluated using magnetic resonance cholangiopancreatography (MRCP).

We performed EUS-FNAB using an Olympus GF-UCT240 linear echoendoscope (Olympus, Tokyo, Japan) and a 22-G needle (NA-11J-KB aspiration needle; Olympus, Tokyo, Japan, EZ shot 3 plus aspiration needle; Olympus, Tokyo, Japan, Expect aspiration needle; Boston Scientific Co., Marlborough, MA, USA, or Acquire aspiration needle; Boston Scientific Co., MA, USA).

Tissue samples were collected by moving the needle around 10–20 times using the slow-pull and funning technique. Tissue samples were assessed for adequacy by performing rapid on-site cytologic evaluation. The number of punctures was determined at the discretion of the surgeon. The EUS-FNAB specimens were processed as described below (Figure 1) [14]:

A piece of tissue aspirated into the needle is pushed onto a glass slide with a stylet;The tubular tissue piece in blood is picked up and transferred to a formalin-soaked plate;The remaining bloody liquid portion is clamped between two glass slides, fixed with ethanol, stained with Papanicolaou stain and submitted as a cytology specimen; andThe removed ductal tissue pieces, which consist of white-toned pancreatic tissue and red-toned pancreatic tissue, are separated using an 18-G disposable needle, transferred separately to a formalin-filled container and submitted to the pathologist as a histopathology specimen.The tissue specimens are quickly fixed in formalin and embedded in paraffin. The paraffin block is cut into thin serial sections and stained with hematoxylin-eosin stain, Masson’s trichrome stain and Elastica–Masson stain. Immune-histochemistry was then performed using IgG4 antibody and cluster of differentiation 38 (CD38) antibody.

Two pathologists evaluated the tissue specimens based on the ICDC histopathological diagnosis using number of high-power fields (HPFs) (×400), number of IgG4-positive cells, GELs (ICDC level 1 histological findings) and granulocytic and lymphoplasmacytic acinar infiltrate (ICDC level 2 histological finding).

## 3. Results

### 3.1. Clinical Findings

Table 1 shows the clinical characteristics of the 10 patients included in this study. The male-to-female ratio was 8:2, and the age (mean ± standard deviation) was 35.6 ± 15.5 years. All the patients (100%) had pancreatic enlargement; five patients (50.0%) had diffuse enlargement, two patients (20.0%) had segmental enlargement, and three patients (30.0%) had focal enlargement. ERCP or MRCP revealed main pancreatic duct narrowing in eight patients (80.0%). Of the eight patients, four (50.0%) had diffuse narrowing, three (37.5%) had segmental narrowing, and one (12.5%) had focal narrowing. The serum IgG4 level of all the patients was <135 mg/dL. All the patients (100%) had UC. Four of the 10 patients (40.0%) had a history of steroid treatment for UC, but none of them received steroids at the stage when AIP was suspected (i.e., at the time of EUS-FNAB).

### 3.2. Histopathological Findings

Table 2 shows the pathological characteristics of the tissue specimens obtained using EUS-FNAB. Eight of the 10 specimens (80.0%) had adequate tissue amounts with HPF > 10. Four of the 10 specimens (40.0%) showed GELs and nine of the 10 specimens (90.0%) showed granulocytic and lymphoplasmacytic acinar infiltrate. Based on the ICDC, the four patients with GELs were diagnosed as confirmed cases of type 2 AIP (Figure 2). Two of the four patients with GELs had lower bile duct stenosis, while the other two patients were asymptomatic. The two symptomatic patients with GELs were treated with steroids and they responded well to treatment. None of the five patients with ICDC level 2 histological findings was on induction of steroids, so a definitive diagnosis based on ICDC was not made.

### 3.3. Progress of Treatment

Regarding the four patients with ICDC level 1 findings, steroid therapy was administered to the two patients with lower bile duct stricture and liver dysfunction. The treatment was effective, and the progress of treatment was consistent with a definitive diagnosis of type 2 AIP. The other two patients were asymptomatic and were not treated with steroids. To date, none of the treated patients have had symptom relapse.

### 3.4. Case Report (Case 10)

A 31-year-old man was diagnosed with left-sided UC and started on remission and maintenance therapy with 5-aminosalicylic acid. He was not administered steroids. Four months after UC diagnosis, he presented to his previous physician with upper abdominal pain and was referred to our hospital due to elevated levels of hepatobiliary enzymes on blood test. The hepatobiliary enzymes with elevated levels include aspartate aminotransferase (45 U/L), alanine aminotransferase (242 U/L), alkaline phosphatase (667 U/L), and γ-glutamyl transpeptidase (785 U/L). Serum IgG4 level was 37 mg/dL. Computed tomography and magnetic resonance imaging showed diffuse pancreatic enlargement and MRCP showed intrapancreatic bile duct stricture and intrahepatic bile duct dilation (Figure 3). ERCP showed main pancreatic duct narrowing and stenosis of the distal bile duct (Figure 3). EUS-FNB of the pancreatic head was performed using a 22-G needle (Acquire aspiration needle; Boston Scientific Co., MA, USA). Histo-pathologically, GELs were observed, but the number of IgG4-positive cells was small at 5–6 cells/HPF and there were no malignant findings (Figure 4). Since the patient had ICDC level 1 imaging and histological findings, he was diagnosed with type 2 AIP. After endoscopic placement of a biliary stent for bile duct stricture, treatment with 40.0 mg of prednisolone was initiated. After tapering the dose to 20.0 mg, the pancreatic swelling and intrapancreatic bile duct stricture improved (Figure 5). The biliary stent was then removed and prednisolone was tapered off, but the patient did not have symptom relapse.

## 4. Discussion

The histopathological diagnostic criteria that constitute the ICDC were based on core biopsy and excisional specimens. In 2011, when the ICDC was published, the histopathological diagnosis of AIP using EUS-FNAB was considered difficult because adequate tissue collection using EUS-FNAB was difficult [22,23,24]. New devices such as Franseen needle and fork-tip needle have recently been introduced and they facilitate adequate tissue collection and demonstrate the usefulness of EUS-FNAB for the diagnosis of AIP and other pancreatic diseases [19,20,25,26,27,28].

Only eight cases of type 2 AIP have been reported in previous studies; in contrast, studies have reported many cases of type 1 AIP [26]. Detlefseen et al. reported two cases of type 2 AIP with ICDC level 1 histopathological findings using EUS-FNB with a 22-G needle [20], and Matsumoto et al. reported a case of type 2 AIP with ICDC level 2 histopathological findings using endoscopic ultrasound-guided fine needle aspiration (EUS-FNA) with a 22-G needle [21], but the usefulness of EUS-FNAB for the histological diagnosis of type 2 AIP was not demonstrated.

The aim of this study was to clarify the usefulness of EUS-FNAB for the histological diagnosis of type 2 AIP. Of the 10 patients, EUS-FNB needle (Acquire aspiration needle; Boston Scientific Co., MA, USA) was used in two cases and EUS-FNA needle (NA-11J-KB aspiration needle; Olympus, Tokyo, Japan, EZ shot 3 plus aspiration needle; Olympus, Tokyo, Japan, Expect aspiration needle; Boston Scientific Co., MA, USA) in eight cases. Regardless of needle type, sufficient tissue samples with HPF > 10 were obtained in 80.0% of cases. Several studies have shown the usefulness of EUS-FNB with a 22-G needle for the histological diagnosis of AIP [14,19,29,30,31]. In this study, a 22-G needle was also used in 9 out of 10 cases, and sufficient tissue amounts were collected. Of the 10 patients, four had GELs corresponding to ICDC level 1 and five had granulocytic and lymphoplasmacytic acinar infiltrate corresponding to ICDC level 2. Of the eight cases with EUS-FNA needles, histopathological findings of ICDC level 1 were obtained in three cases (37.5%), histopathological findings of ICDC level 1 or 2 findings in eight cases (100%); of the two cases with EUS-FNB needles, histopathological findings of ICDC level 1 were obtained in 1 case (50.0%). Yoon et al. reported that histopathological findings of ICDC level 1 were obtained in 40.0% of cases with EUS-FNA and level 1 or 2 results in 80.0% (three studies, 10 patients), and with EUS-FNB excluding the 19-gauge Trucut needle, level 1 results were obtained in 33.3% and level 1 or 2 results in 66.7% (two studies, three patients) [26]. The results of this study are considered consistent with this systematic review. EUS-FNB can be considered a useful tissue collection method for the histological diagnosis of type 2 AIP.

This study has several limitations. First, it was a single-center retrospective observational study. Second, the number of patients in the study was small. Third, some patients had already received steroids prior to EUS-FNAB, and the steroids may have masked the inflammation. Fourth, it may be difficult to distinguish AIP from non-specific pancreatitis in patients with UC.

Although the usefulness of EUS-FNB for the histological diagnosis of type 1 AIP has been reported in previous studies, pathological diagnosis based on EUS-FNB specimens requires a multidisciplinary approach involving collaboration between clinicians, radiologists and pathologists [24,26,32,33]. Although EUS-FNB with the newly developed devices has made collection of sufficient tissue amounts possible, it has been reported that some cases cannot be diagnosed as type 1 AIP even if the tissue amount is sufficient [13]. In addition, pathologists often disagree on the diagnosis and histological diagnosis based on EUS-FNB specimens has not been established, even for type 1 AIP [32]. According to a recent meta-analysis, the diagnostics yield for level 1 or 2 histology criteria of type 1 AIP was 87.2% for EUS-FNB, and the yield for level 1 was 60.1%. Although GELs are a characteristic finding in type 2 AIP, the glandular ducts obtained from EUS-FNAB specimens are often small and diagnosis based on these tissue specimens is controversial. The findings of this study suggest that EUS-FNAB can provide sufficient tissue samples, but further large-scale studies on the histopathological diagnosis of type 2 AIP based on EUS-FNAB specimens are needed.

## 5. Conclusions

EUS-FNAB is a useful minimally invasive alternative to resection and core-needle biopsy for the diagnosis of type 2 AIP because it provides sufficient evaluable tissue samples.

## Figures and Tables

**Figure 1 diagnostics-12-02464-f001:**
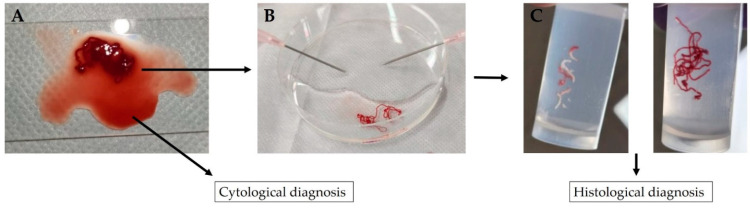
Processing of histological specimens. (**A**) Tubular tissue fragments and bloody liquid components extruded onto a glass slide. (**B**) Ductal tissue pieces separated into white-toned and red-toned pancreatic tissues using 18-G disposable needles. (**C**) White-toned and red-toned pancreatic tissues transferred into separate formalin-filled containers.

**Figure 2 diagnostics-12-02464-f002:**
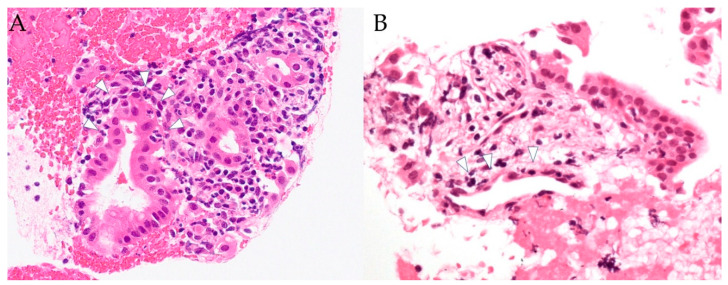
Histopathological tissue obtained using EUS-FNB from a patient with a confirmed diagnosis of type 2 autoimmune pancreatitis. (**A**,**B**) Histological examination of specimens showing GELs (arrowhead, hematoxylin–eosin stain, ×40).

**Figure 3 diagnostics-12-02464-f003:**
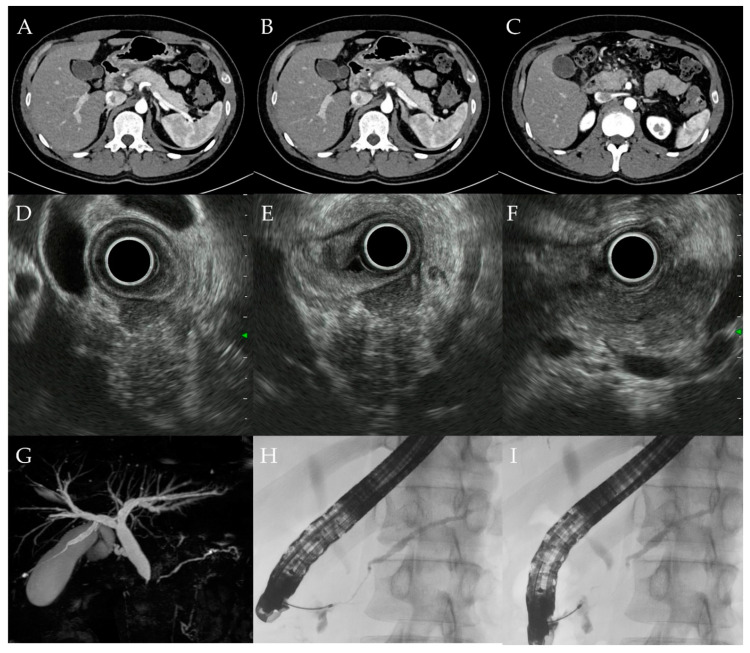
Image findings of case 10. (**A**–**C**) Computed tomography (CT) showing diffuse pancreatic enlargement. (**D**–**F**) Endoscopic ultrasound showing diffuse pancreatic enlargement and a mass in the pancreatic head. (**G**) Magnetic resonance imaging showing intrapancreatic bile duct stenosis and MPD narrowing. (**H**,**I**) Endoscopic retrograde cholangiopancreatography (ERCP) showing MPD narrowing and intrapancreatic bile duct stenosis.

**Figure 4 diagnostics-12-02464-f004:**
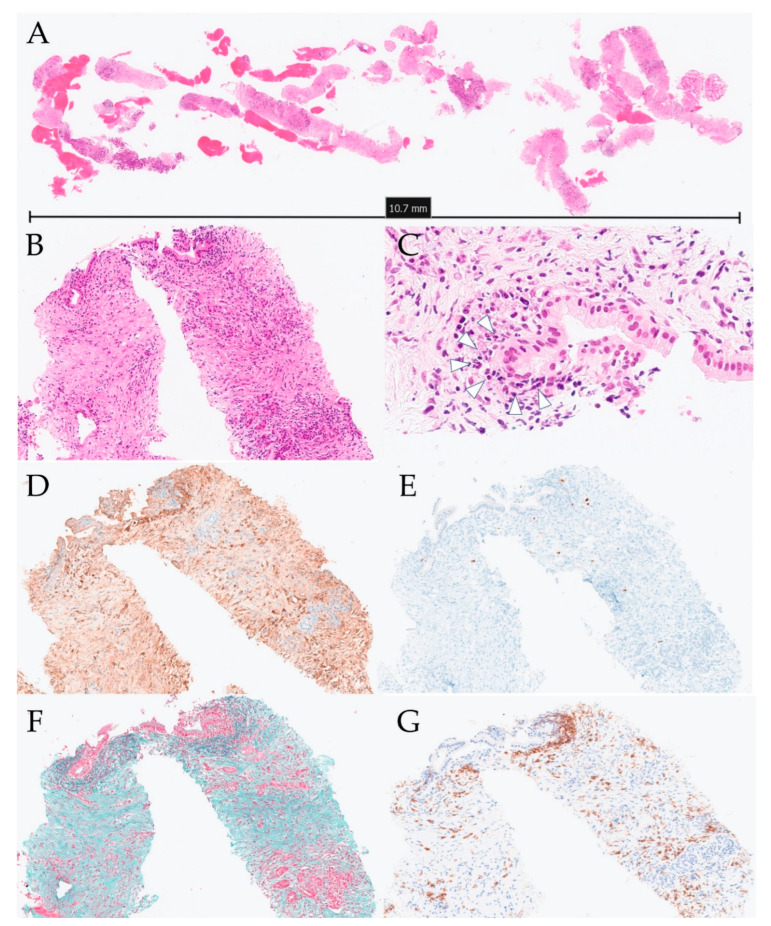
Histopathological tissue from case 10 obtained using EUS-FNB. (**A**) Loupe image of specimen obtained using EUS-FNB showing adequate amount of tissue (hematoxylin–eosin stain) (**B**) Microscopic image showing neutrophilic infiltration of pancreatic parenchyma (hematoxylin–eosin stain, ×10). (**C**) Microscopic image showing GELs (arrowhead, hematoxylin–eosin stain, ×40). (**D**) IgG immunostaining showing IgG-positive cells (×10). (**E**) IgG4 immunostaining showing a few IgG4-positive cells (×10). (**F**) Elastica–Masson staining showing no obstructive phlebitis (×10). (**G**) Cluster of differentiation 38 immunostaining showing plasma cells (×10).

**Figure 5 diagnostics-12-02464-f005:**
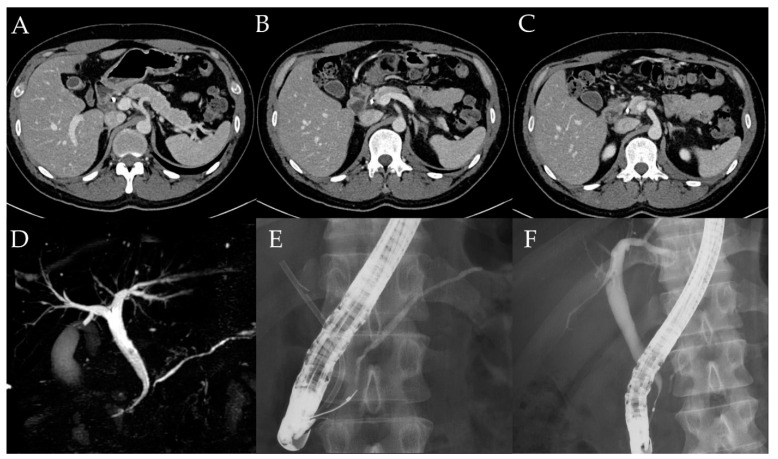
Imaging findings after steroid introduction. (**A**–**C**) CT. (**D**) Magnetic resonance cholangiopancreatography. (**E**,**F**) ERCP. Pancreatic enlargement, MPD stenosis, and bile duct stricture all improved after steroid introduction.

**Table 1 diagnostics-12-02464-t001:** Clinical characteristics of patients.

Characteristic	Value	Characteristic	Value
Sex, male-to-female ratio	8:2	Age (mean ± SD), years	35.6 ± 15.5
Symptom		Pancreatic imaging finding	
Abdominal pain, n (%)	4/10 (40.0%)	Enlargement, n (%)	10/10 (100%)
Jaundice, n (%)	2/10 (20.0%)	Diffuse enlargement, n (%)	5/10 (50.0%)
Asymptomatic, n (%)	4/10 (40.0%)	Segmental enlargement, n (%)	2/10 (20.0%)
Serology variable		Focal enlargement, n (%)	3/10 (30.0%)
IgG4 level (mean ± SD), mg/dL	34.9 ± 17.3	MPD narrowing, n (%)	8/10 (80.0%)
IgG4 level < 135 mg/dL	10/10 (100%)	Diffuse narrowing, n (%)	4/8 (50.0%)
OOI		Segmental narrowing, n (%)	3/8 (37.5%)
IBD (ulcerative colitis), n (%)	10/10 (100%)	Focal narrowing, n (%)	1/8 (12.5%)

IBD, inflammatory bowel disease; IgG, immunoglobulin G; MPD, main pancreatic duct; OOI, other organ involvement; SD, standard deviation.

**Table 2 diagnostics-12-02464-t002:** Histopathological findings.

Case	Sex/ Age, Years	Needle (Gauge)	HPF	IgG4/HPF	GEL	Granulocytic and Lymphoplasmacytic Acinar Infiltrate	Diagnosis (ICDC)	Steroid Administration (Response to Steroid)
1	M/37	AC (22-G)	>10	absent	−	−	n/d	−
2	F/49	EX (22-G)	>10	scant	+	+	Level 1	−
3	M/30	NA (22-G)	6	absent	−	+	Level 2	−
4	M/68	EX (19-G)	>10	scant	−	+	Level 2	−
5	M/21	EX (22-G)	>10	scant	+	+	Level 1	−
6	M/53	EZ (22-G)	>10	scant	−	+	Level 2	−
7	M/29	NA (22-G)	>10	scant	+	+	Level 1	+ (good)
8	M/18	NA (22-G)	4	absent	−	+	Level 2	−
9	F/20	NA (22-G)	>10	scant	−	+	Level 2	−
10	M/31	AC (22-G)	>10	scant	+	+	Level 1	+ (good)

AC, Acquire aspiration needle; EX, Expect aspiration needle; EZ, EZ shot 3 plus aspiration needle; F, female; GEL, granulocytic epithelial lesion; HPF, high-power field; ICDC, International Consensus Diagnostic Criteria; M, male; NA, NA-11J-KB aspiration needle; n/d, not determined.

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
