# Peer review of "Utility of Endoscopic Ultrasound-Guided Fine-Needle Aspiration and Biopsy for Histological Diagnosis of Type 2 Autoimmune Pancreatitis"

_diagnostics, 2022, doi:10.3390/diagnostics12102464_

Round 1

Reviewer 1 Report

The authors performed a retrospective study to clarify the usefulness of EUS-guided FNB for the diagnosis of type 2 AIP. The manuscript was interesting, and I have only several minor comments.

1. The authors reported their experience of EUS-FNB for ten patients with type 2 AIP. This is relatively many patients, given the rarity of this entity. I think this manuscript can prove valuable for the readership of “Diagnostics.”

2. I do not fully agree with the authors’ statement that “Although the usefulness of EUS-FNB for the histological diagnosis of type 1 AIP has been reported in previous studies, pathological diagnosis based on EUS-FNB specimens remains controversial.” According to a recent meta-analysis (Yoon S, et al. DEN 2021;33:1024-1033), the diagnostic yield for level 1 or 2 histology criteria of AIP was 87.2% for FNB, and the yield for level 1 histology was 60.1% for FNB. EUS-FNB may provide a pivotal role in the diagnosis of type 1 AIP, although a multidisciplinary approach with the collaboration of clinicians, radiologists, and pathologists may be necessary to make a diagnosis of AIP.

3. According to Table S1 of a systematic review (DEN 2021;33:1024-1033), a non-weighted average of level 1 histology for type 2 AIP (4 studies, 15 patients) was 67% (10/15) for FNB, and a non-weighted average of level 1 or 2 histology for type 2 AIP was 87% (13/15) for FNB. Comparing the authors’ data with this data from a systematic review would be more informative.

Author Response

Thank you for the thoughtful and constructive feedback you have provided regarding our manuscript.  We have addressed your comments with point-by-point responses and revised the manuscript accordingly.

Our replies to the comments from Reviewer 1

Reviewer: 1

  1. The authors reported their experience of EUS-FNB for ten patients with type 2 AIP. This is relatively many patients, given the rarity of this entity. I think this manuscript can prove valuable for the readership of “Diagnostics.”

We appreciate your assessment. We are also glad that your remarks will help us improve our paper.

  1. I do not fully agree with the authors’ statement that “Although the usefulness of EUS-FNB for the histological diagnosis of type 1 AIP has been reported in previous studies, pathological diagnosis based on EUS-FNB specimens remains controversial.” According to a recent meta-analysis (Yoon S, et al. DEN 2021;33:1024-1033), the diagnostic yield for level 1 or 2 histology criteria of AIP was 87.2% for FNB, and the yield for level 1 histology was 60.1% for FNB. EUS-FNB may provide a pivotal role in the diagnosis of type 1 AIP, although a multidisciplinary approach with the collaboration of clinicians, radiologists, and pathologists may be necessary to make a diagnosis of AIP.

We have revised the discussion by citing the results of the meta-analysis you provided (Page8, line 196-199).

  1. According to Table S1 of a systematic review (DEN 2021;33:1024-1033), a non-weighted average of level 1 histology for type 2 AIP (4 studies, 15 patients) was 67% (10/15) for FNB, and a non-weighted average of level 1 or 2 histology for type 2 AIP was 87% (13/15) for FNB. Comparing the authors’ data with this data from a systematic review would be more informative.

We have revised the discussion as you have indicated. We have listed the results by needle used so that the results can be compared with the results of the systematic review (Page9, line219-226).

Reviewer 2 Report

The paper is interesting and there's scanty data on EUS-FNB of AIP-2. I have the following comments:

1) Although the title specifies the use of FNB, some patients were sampled with FNA needles (material and methods section). The authors should clarify this aspect and they should also specify which needle was used in each patients in Table 2.

2) English grammar should be improved

3) The limited number of patients is a limitation to the paper, although the current reviewer acknowledge the fact that AIP-2 is a very rare condition

4) The authors should enrich the discussion commenting the current of art on EUS-guided tissue sampling of pancreas (cite the recent meta-analyses PMID: 35124072 PMID: 32014422)

Author Response

Thank you for the thoughtful and constructive feedback you have provided regarding our manuscript.  We have addressed your comments with point-by-point responses and revised the manuscript accordingly.

Our replies to the comments from Reviewer 2

Reviewer:2

1) Although the title specifies the use of FNB, some patients were sampled with FNA needles (material and methods section). The authors should clarify this aspect and they should also specify which needle was used in each patients in Table 2.

Table 2 lists the needles used in each case. As noted by you, this retrospective study reported mixed results for both FNA and FNB needles. Therefore, we are correcting the results to EUS-FNAB results. The number of cases with FNA and FNB needles is also listed on page 8-9, line 209-213.

2) English grammar should be improved

Please forgive our poor English writing. Although we are not native English speakers, The manuscript has been carefully reviewed by an experienced editor whose first language is English and who specializes in editing papers written by scientists whose native language is not English. This information is added in the acknowledgements (Page10, line269). A certificate of proofreading in English is attached.

3) The limited number of patients is a limitation to the paper, although the current reviewer acknowledge the fact that AIP-2 is a very rare condition

The small number of patients is noted as a limitation of this study (Page9, line 230-234).

4) The authors should enrich the discussion commenting the current of art on EUS-guided tissue sampling of pancreas (cite the recent meta-analyses PMID: 35124072 PMID: 32014422)

The same information has been pointed out by other reviewers. We have revised our discussion based on the results of this meta-analysis(Page9, line 209-250).

Round 2

Reviewer 2 Report

The manuscript is OK now. Thank you!